# Trends of community-based systemic antibiotic consumption: Comparative analyses of data from Ethiopia and Norway calls for public health policy actions

Girma Gutema[1,2]*, Seid Ali[3], Sultan Suleman[4]

1 Institute of Pharmaceutical Sciences, University of Oslo, Oslo, Norway, 2 Faculty of Health Sciences, Rift Valley University, Adama, Oromia, Ethiopia, 3 Swiss Tropical and Public Health Institute, University of Basel, Basel, Switzerland, 4 Department of Pharmaceutical Analysis and Regulatory Affairs, Jimma University, Jimma, Oromia, Ethiopia

* girmabbaacabsaa@gmail.com, girm2004@yahoo.com

**Data Availability Statement:** Relevant data are within the paper and its Supporting information file.

**Funding:** The authors received no specific funding for this work.

## Abstract

Studies on antibiotic utilization trends are invaluable because they offer data for evaluation of impacts of antimicrobial stewardship policies. Such studies help determine correlations between the use of specific antibiotic classes and trends in emergence of resistance (resistance-epidemiology). This study aims to quantify the consumption systemic antibiotics (J01)—in defined daily doses (DDD) per 1000 inhabitants per day (DID)—in Ethiopia's public healthcare sector (2016–2020). By so doing, it attempts to capture the extent of population exposure to antibiotics in the country. Data were also compared with those from Norway to establish rough estimate of the country's status vis-à-vis some globally acknowledged better practices with regard to optimal use of antibiotics. Raw data obtained from registers of Ethiopian Pharmaceutical Supply Agency were converted into DDD, per the standard methodology recommended by WHO. To control for population size, antibiotics consumption data were presented as DID. Since official population census data for Ethiopia were not available for the study period, population projection data from the World Bank were used. Community-based consumption of systemic antibiotics increased from 11.02 DID in 2016 to 12.83 DID in 2020 in Ethiopia—an increase by 16.4%. Moreover, analysis of a log-linear regression model showed that the average growth rate in the community-based systemic antibiotics consumption per year between 2016 and 2020 was about 3.3% ($R^2$ = 0.89). The highest percentage change in community-based systemic antibiotics consumption happened for glycopeptides (J01XA) and the fourth generation cephalosporins (J01DE)—1300% and 600% compared to the baseline year (2016), respectively. At product level, 9 antibiotics constituted the common domain in the list of medication cocktails in the drug utilization 90% (DU90%) for the study period. Community-based consumption of systemic antibiotics for Ethiopia and Norway showed opposite trends, calling for public health policy actions in Ethiopia.

**Competing interests:** The authors have declared that no competing interests exist.

## Introduction

Antibiotics, group of drugs which treat or prevent spread of infections by either inhibiting the growth and replication of bacteria or outrightly killing them, are among the most widely used classes of medications both in human and animal health sectors. The associations between the use of increasing quantities of antibiotics and the development of resistance have been well documented for both nosocomial and community-acquired infections in various parts of the world [1–5]. Such associations also hold true for livestock sector where antibiotics are used for the purposes of therapy and/or growth promotion (food additives) in animals [6–8].

Consequently, selective pressure exerted by widespread antibiotic use has become the major determinant factor for the development of drug resistance across different sectors in human health, veterinary medicine and animal husbandry. Moreover, resistance factors in human and animal pathogens and commensals, particularly those carried on mobile genes, can spread rapidly within as well as between human and animal populations in the wider eco-system they share [9, 10]. As resistance develops as a function of complex correlation to the quantities of antibiotics consumed in anyone or all of such sectors, its emergence and spread can be prevented primarily by limiting the irrational use of antibiotics [11].

What's more, antibiotics constitute a very unique class of drugs in the medical catalogue in that their effectiveness decreases—due to the natural and evolutionarily driven process called resistance—as a function of the extent to which they are being utilized [12]. As a result of the various driving factors including tourism/medical tourism, migration, food chains, international trade and generally globalization, resistance has become a global challenge and hence any meaningful effort to tackle this challenge needs concerted global collaborations [13, 14]. Moreover, antibiotics now hold a unique position in the medical catalogue not just because they are the only drugs which lose effectiveness with increasing utilization, but they also are considered "societal drugs" for their rational use or misuse can benefit or harm patients, anywhere in the world, including those who did not receive them for any medical purposes before [15].

In Ethiopia, comprehensive surveillance data on antibiotic use and resistance at national level are yet lacking. In May 2018, a practical guide to establish antimicrobial stewardship program in all hospitals across the country was released [16]. The proposed program included surveillance of antibiotic consumption in addition to the detection and reporting of resistance. No trend data from this surveillance system have been made publicly available yet. A few isolated studies which documented either antibiotic consumption or resistance at the level of specific healthcare facilities have however been published [17–19]. One of these studies reported average annual consumption of systemic antibiotics in three clinical wards at the country's largest hospital to be 81.6DDDs/100 bed days [19]. Besides, systematic review of antibiotic resistance data from some African countries, including Ethiopia, was reported elsewhere [20]. On the other hand, countries like Norway have maintained trend data on consumption of antibiotics since the mid-1970s. For instance, the NORM/NORM-Vet 2016 report showed an increasing trend of consumption for penicillins (J01C) from 1974 to 2012 in Norway—from about 3.5 to 8.5DDDs/1000 inhabitants/day. However, a declining trend for such community-based consumption for sulfonamides and trimethoprim (J01E) had also been shown in the report [21].

Literature is scarce on the public health and economic burdens of antibiotic resistant infections particularly in low and middle income countries. In the United States, however, over 2 million people are estimated to contract antibiotic resistant infections, of whom over 23,000 die of such infections every year [22]. In Europe, about same mortality rate has been reported yet with a far less morbidity rate as compared to those of the US—about 400,000 people

contract antibiotic resistant infections, of whom over 25,000 die every year [23]. Moreover, most of the morbidities and mortalities in patients receiving surgical interventions are directly related to antibiotic resistant infections in hospital settings [24]. Besides, the economic burden due to antibiotic resistant infections is about 1.5 billion Euros per year in Europe, about 90% of which is said to correspond to direct hospital costs [23]. In the US, the overall economic costs of antibiotic resistant infections are estimated to be about 55 billion dollars per year, about 36% of which accounting for direct healthcare costs and the rest related to the productivity lost due to morbidities associated with antibiotic resistant infections [22]. For all such heavy impacts of antibiotic resistant infections, be that in the public health or economic sectors, over use and/or inappropriate use (irrational use) of antibiotics remain to be the key drivers [25].

## Significance of the study

In many countries, especially those with well-developed healthcare systems, comprehensive data on the annual consumption of all drugs in human medicine, or even in other sectors like veterinary, animal husbandry or aquaculture, are published on regular basis by healthcare authorities [26]. These data are often presented in standardized measurement units like the number of defined daily doses (DDD) per 1000 inhabitants per day (DID) for the systematic documentation and control of antibiotics consumption trends in healthcare settings.

Such standardized measurements enable researchers to compare their findings from different countries. More importantly, such an approach makes it possible—by ensuring availability of the harmonized data sets needed—that concerted international efforts can be made to tackle the overuse or misuse of antibiotics that drive the global challenge of antibiotic resistance. And particularly for antibacterial drugs, such data sets provide the basis for the evaluation of therapeutic trends in bacterial infections and human exposure to antibiotics [27].

Data on annual antibiotic consumptions in public health sector are not publicly available for Ethiopia, as is often the case for most other low and middle income countries. Such data for other sectors in which antibiotics are widely used including in veterinary medicine and poultry farming are also lacking. What's more, both quantitative and qualitative records of antibiotic utilization trends are essential, for not only they enable healthcare authorities evaluate the impacts of antimicrobial policy in a country, but also because such data allow the determination of strong correlations between the use of certain specific groups of antibiotics and the emerging trends of antibiotic resistance—commonly called resistance-epidemiology [28, 29].

To date, no study has been conducted in the Ethiopian healthcare systems, whether in public or private sectors, on the trends of annual systemic antibiotics (J01) consumptions at country level. This study, therefore, aimed at quantifying the annual consumption systemic antibiotics (J01) for human use—in defined daily doses per 1000 inhabitants per day (DID)—in Ethiopia's public healthcare sector, from 2016–2020, thereby attempting to estimate the extent of population exposure to antibiotics in the country during the stated study period. Moreover, trend data for Ethiopia were also compared with those from Norway to establish some rough estimate of the country's general status vis-à-vis the globally acknowledged better practices with regard to controlled and optimal utilization of systemic antibiotics at community levels.

## Materials and methods

Data from the registers of the Ethiopian Pharmaceutical Supply Agency (EPSA) were collected and used in the current study. EPSA was established in 2007, with the parliamentary

proclamation number: 553/2007—then with another name "Pharmaceutical Fund and Supply Agency, PFSA"—as an autonomous organ of the federal government accountable to the Ministry of Health. Headquartered in the capital Addis Ababa, EPSA serves as the central coordinating hub for the procurement, storage and distribution of essential pharmaceuticals needed across Ethiopia's public health sector with three legislatively stipulated objectives: 1- enabling public health institutions to supply quality assured essential pharmaceuticals at affordable prices in a sustainable manner to the public; 2- playing a complementary role in developmental efforts for health service expansion and strengthening, by ensuring enhanced and sustainable supply of pharmaceuticals and 3- creating conditions which enable accumulation of funds in its revolving and cost recovery processes thereby ensuring the realization of the above two objectives [30].

As per the official sources, EPSA's administrative structure encompasses multiple internal departments each with defined responsibilities. These include Planning Monitoring and MIS Directorate, Storage and Distribution Directorate, Fund Management Directorate, Forecasting and Capacity Building Directorate, Pharmaceuticals and Medical Supplies Procurement Directorate, and Human Resource and General Service Directorate. In addition to its head quarter situated at the Gulale Sub-city of the capital Addis, EPSA has got 19 branches grouped into 7 "cluster" across the 9 regional states and two administrative city councils in Ethiopia's federal arrangement. The clusters and branches serve as distribution outposts for medicines, laboratory reagents, medical supplies and equipment procured by the head office to localities all over the country [31].

## Study design

The design of this study is institution-based retrospective cross-sectional type.

## Source population

All medications supplied by the central EPSA to its branch offices during the study period [2016–2020]

## Study population

All antibiotics for systemic use (Anatomic Therapeutic Chemical J01) supplied by the central EPSA to all its branch offices during the period.

## Exclusion criteria

Antibiotics in pediatric formulations, dermatologicals (topical products), anti-tuberculotic drugs as well as other antibiotics for eye and ear treatments supplied by the central EPSA to its branch offices during the study period were excluded—this being a methodological requirement.

## Data collection and management

**Sample size.** This is a whole population study and hence all the systemic antibiotics used each year during 2016–2020 in Ethiopia's public health sector were quantified and analyzed in the current study.

**Source of data.** Forecasting and quantification data, as reconciled with distribution/delivery registers from the central EPSA warehouses to all its regional branches were used to collect the data. Community-based systemic antibiotic consumption were therefore computed based on central EPSA's comprehensive data registers. Accordingly, the agency's statistical records

on unit quantities such as packs, boxes (number of tablets/capsules), vials, ampoules, MIU etc, of systemic antibiotics supplied to the different branches/regional hubs—split according to their routes of administration—were recorded on a pre-piloted and validated paper-based data collection format. The raw data were then converted into Defined Daily Doses (DDD) according to the Anatomical Therapeutic Chemical classification system (ATC/DDD), so that making standard comparisons on drug utilization review as well as rough estimations on per capita consumption or population-based exposure to antibiotics can be possible with other countries. To control for population size, consumption data were presented as the number of DDD per one thousand inhabitants per day (DID). Since official population census data for Ethiopia were not available for the study period (2016–2020), population projections data at the World Bank's portal [32] were used for each of the years.

Finally, to compare the annual systemic antibiotics consumption between Ethiopia and Norway through the standardized matrices of DID, data for Norway were collected from the publicly available European Surveillance of Antimicrobial Consumption Network (ESAC-Net) database [33]. Duration of the study period for the comparison of community-based systemic antibiotics consumptions in Ethiopia and Norway was limited by the available data by the time of data collection.

**Data entry and analysis.** Epi Info™, Version 5.5.2 (Center for Surveillance, Epidemiology and Laboratory Services, USA) was used for data entry. The data were then exported to SPSS Statistics, Version 26 (SPSS Inc., Chicago, IL, USA) for statistical analyses. Quantities of DDD (and then DID) for all the systemic antibiotic products were computed based on WHO's Collaboration Center for Drug Statistics Methodology guideline, 2019 edition [34]. Accordingly, DDD of a drug is a standard unit defined based on the average daily dose used for its main indication. For antibiotic products marketed both alone and in the form of fixed-dose combinations (example: Trimethoprims and Sulphonamides), amounts of individual compounds were accounted for separately. For antibiotics combined with beta-Lactamase inhibitors, only amounts of antibiotics were included in the calculations.

What's more, community-based consumptions of the various systemic antibiotics were then ranked according to the specific drug products constituting 90% of the total consumption for each year during the study period—referred to as DU90% [35]. Results presented in table as well as figures, and p-value that does not exceed 0.05 was taken as significant in our statistical analysis.

## Results

Community-based consumption of systemic antibiotics (J01) increased from 11.02 DID in 2016 to 12.83 DID in 2020 in Ethiopia. Analysis of log-linear regression model (see S1 Fig) showed that the average annual growth rate in the community-based systemic antibiotics consumption between 2016 and 2020 was 3.4% ($R^2$ = 0.89). The lowest and the highest increments in consumption however happened during the periods between 2017 to 2018 (-0.3%) and 2019 to 2020 (7.5%), respectively. Furthermore, a closer look at the dis-aggregated data for trend analyses showed that sulfonamides and trimethoprim (J01E) did constitute the highest average increase (10.1%) in community-based systemic antibiotic consumption for each year from 2016–2020 in Ethiopia. In the current study, the lowest figure was however documented for amphenicols (J01B) during same study period (-20.8%). Others like quinolones (J01M), aminoglycosides (J01G) and macrolides (J01F) also showed significant upward spikes in community-based consumption during same periods within the Ethiopian public health sector (Table 1).

**Table 1. Community-based annual consumption of systemic antibiotics (J01) in public health sector in Ethiopia, 2016–2020.**

| Systemic antibiotics (J01) | Community-based annual consumption, DID (%) | | | | |
|---|---|---|---|---|---|
| | **2016** | **2017** | **2018** | **2019** | **2020** |
| Beta-lactam antibiotics, penicillins (J01C) | 4.21 (38.3) | 5.09 (43.8) | 4.61 (39.9) | 4.61 (38.6) | 4.47 (34.8) |
| Tetracyclines (J01A) | 1.86 (16.9) | 1.78 (15.4) | 2.21 (19.1) | 1.66 (13.9) | 2.18 (17) |
| Quinolones (J01M) | 1.55 (14.1) | 1.32 (11.4) | 1.29 (11.2) | 1.26 (10.6) | 2.19 (17.1) |
| Sulfonamides & trimethoprim (J01E) | 1.55 (14.1) | 1.34 (11.6) | 1.18 (10.2) | 2.4 (20.1) | 1.89 (14.7) |
| Other antimicrobials (J01X) | 1.04 (9.5) | 1.04 (9) | 1.24 (10.7) | 1.05 (8.8) | 1.03 (8.1) |
| Macrolides, lincosamides (J01F) | 0.45 (4.1) | 0.59 (5.1) | 0.59 (5.1) | 0.57 (4.8) | 0.64 (5) |
| Other beta-lactams (J01D) | 0.2 (1.8) | 0.28 (2.4) | 0.31 (2.7) | 0.3 (2.5) | 0.31 (2.4) |
| Aminoglycosides (J01G) | 0.12 (1) | 0.13 (1.1) | 0.11 (0.9) | 0.07 (0.5) | 0.11 (0.9) |
| Amphenicols (J01B) | 0.03 (0.3) | 0.03 (0.3) | 0.02 (0.2) | 0.02 (0.2) | 0.01 (0.1) |
| **Total** | **11.02 (100)** | **11.6 (100)** | **11.56 (100)** | **11.94 (100)** | **12.83 (100)** |

The highest percentage change in community-based systemic antibiotics consumption happened for glycopeptides (J01XA) and the fourth generation cephalosporins (J01DE). They showed an increase of 1300% and 600% compared to the baseline year (2016), respectively. On the other hand, amphenicols (J01BA), beta-lactate-sensitive penicillins (J01CE) and intermediate-acting sulfonamides (J01EC) showed quantitatively significant decrease (-68%, -44% and -41%, respectively) in community-based consumption compared to the baseline year (Fig 1).

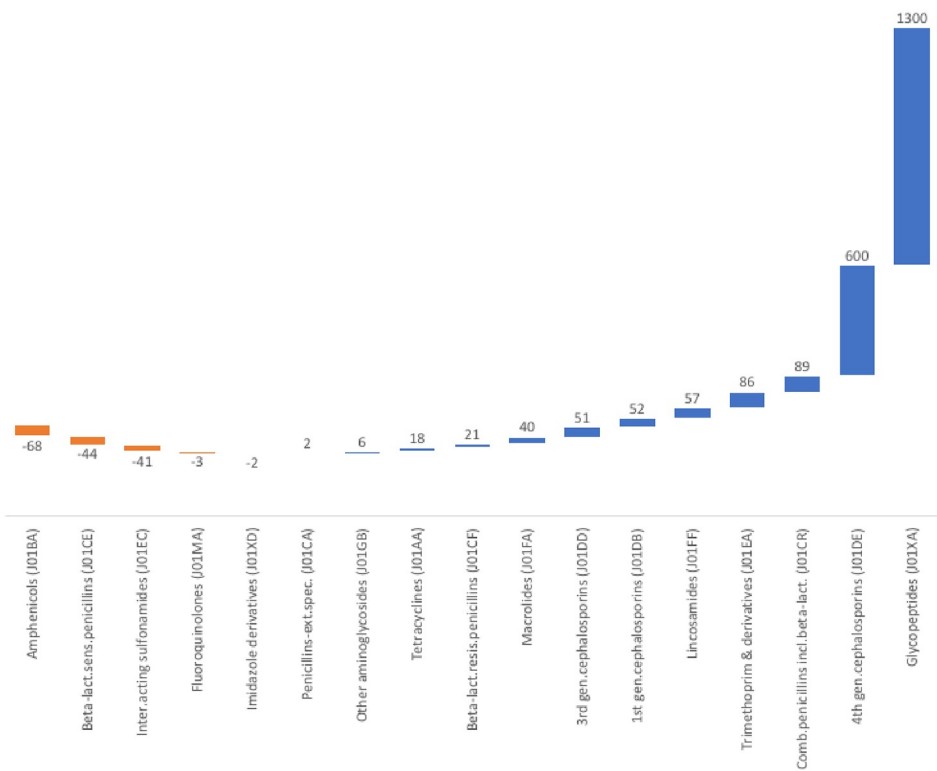

**Fig 1. Percentage changes over baseline in community-based consumption of systemic antibiotics (J01) in Ethiopia between 2016 and 2020.**

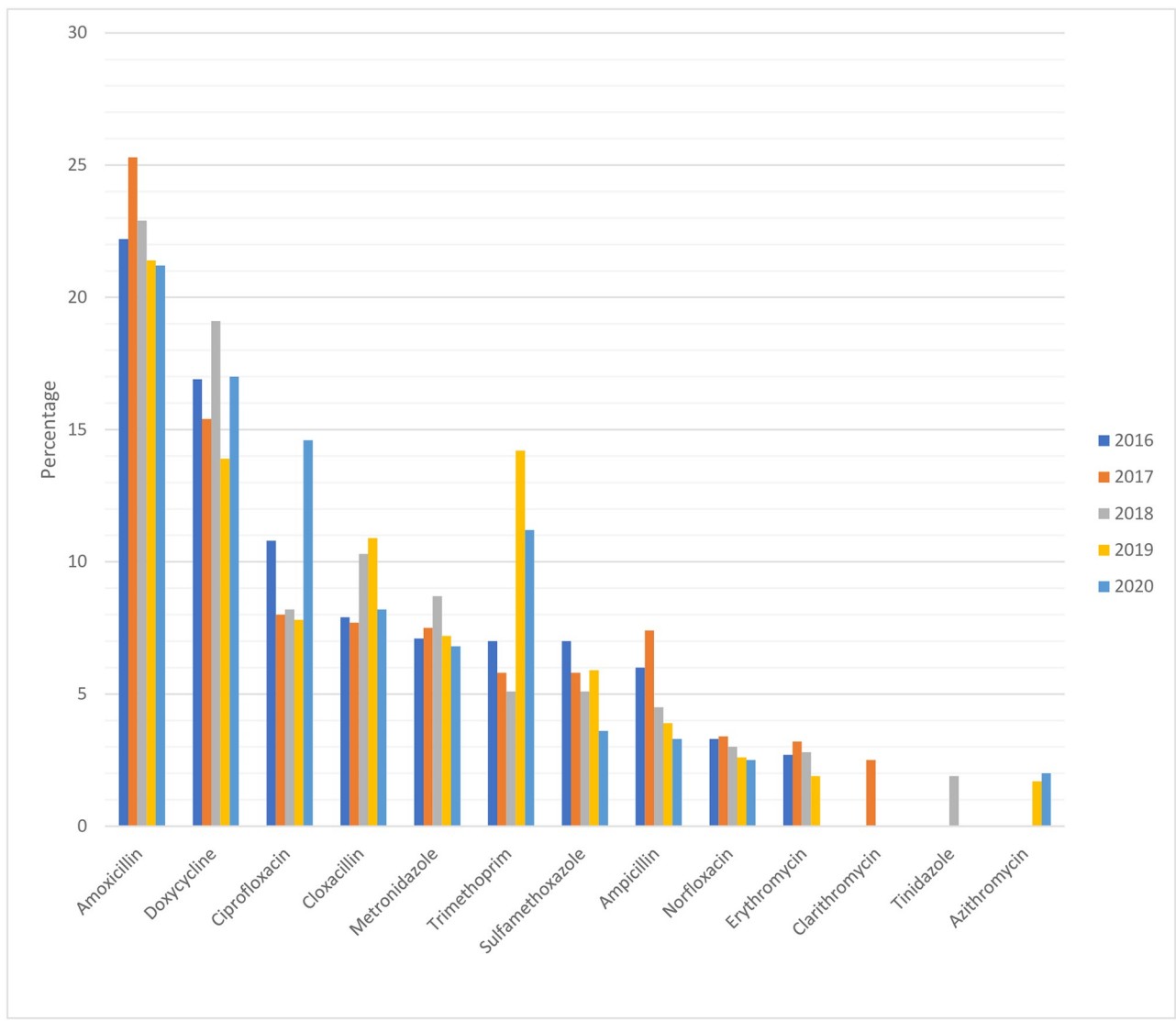

**Fig 2. Products constituting DU90% of the annual community-based systemic antibiotics (J01) consumption in Ethiopia, 2016–2020.**

Ten antibiotic products constituted the drug utilization ninety percent (DU90%) each for 2016 and 2020, and 11 products constituted the DU90% each for the years of 2017, 2018 and 2019. Community-based consumption of amoxicillin topped the DU90% for all the five years of the study period, accounting for 22.2% in 2016, 25.3% in 2017, 22.9% in 2018, 21.4% in 2019 and 21.2% in 2020. Moreover, 9 antibiotic products constituted the common domain in the list of product cocktails in the DU90% for the study period. These products are amoxicillin, doxycycline, ciprofloxacin, cloxacillin, metronidazole, trimethoprim, sulfamethoxazole, ampicillin and norfloxacillin (Fig 2).

Community-based annual consumption of systemic antibiotics in Ethiopia and Norway were, 11DID and 16.2DID in 2016, 11.6DID and 15.8DID in 2017 and, 11.6DID and 15.3DID in 2018, respectively (Fig 3).

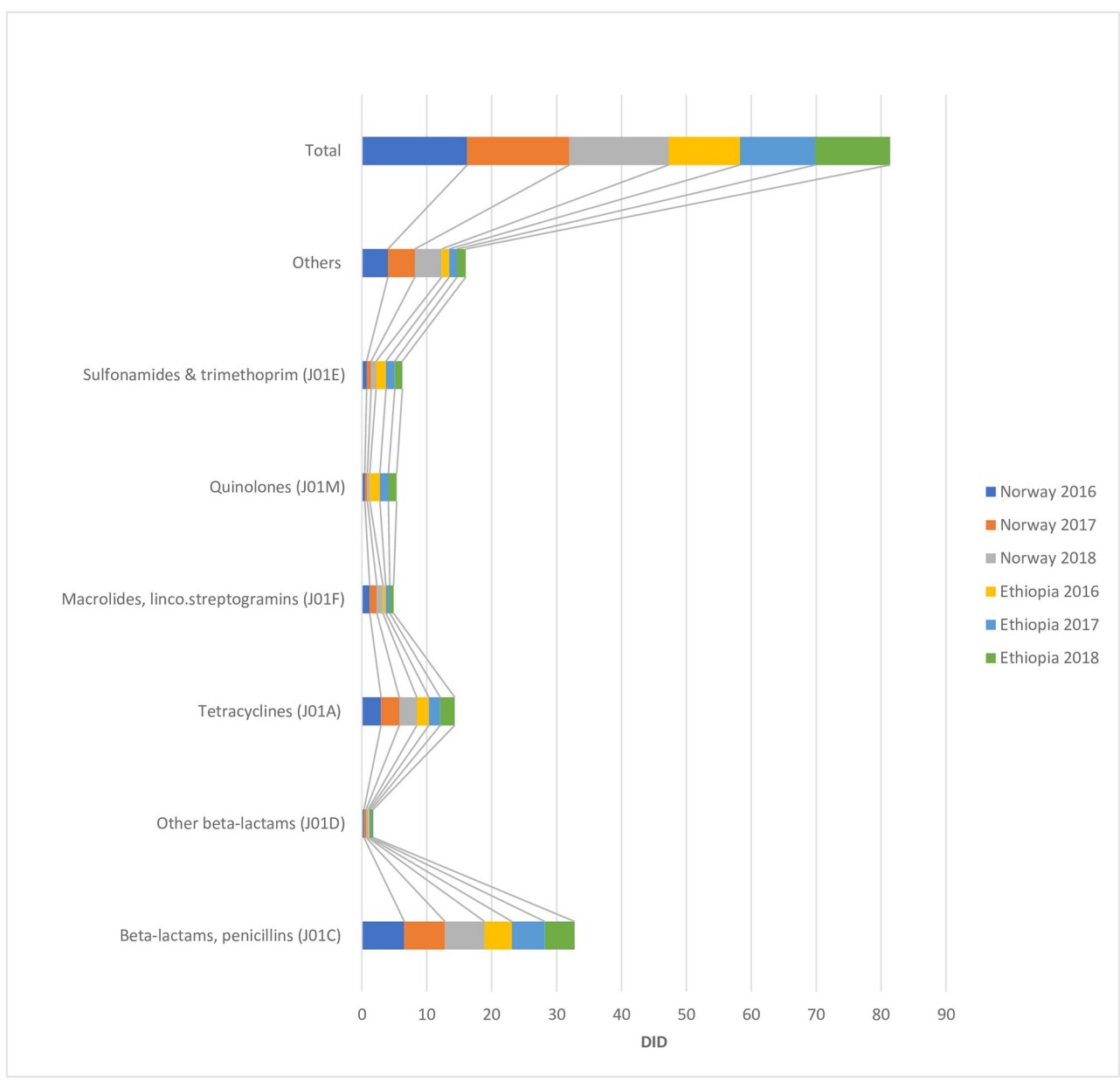

**Fig 3. Comparison of community-based systemic antibiotics (J01) consumption in Ethiopia and Norway, 2016–2018.**

## Discussion

The increase in the community-based consumption of systemic antibiotics from 2016 to 2020 in Ethiopia was quantitatively significant as it did account for an overall percentage rise of 16.4. Moreover, as can be seen from the analysis of the log-linear regression model, the independent variable (time in year) was clearly a significant factor–in fact a driving force–for the variations that occurred in the dependent variable (community-based consumption of systemic antibiotics in DID) because about 89% of the changes that happened to the later did happen due to changes in the former ($R^2 = 0.89$).

Among the different classes of systemic antibiotics, sulfonamides and trimethoprim (J01E) showed a notably large increase, in percentage, for each of the years during the study period (2016–2020), with an average annual percentage increase rate of about three folds of that of the average annual increase for the cumulative consumption of all antibiotics considered in the current study (10.1% Vs 3.4%). In line with this, such increasing consumption of sulfonamide and trimethoprim over earlier years had been the cause of serious concerns elsewhere [36], and that still remains a challenge even in some parts of the world with better developed health-care systems [37].

Concerning the percentage changes in the annual community-based systemic antibiotic consumption observed in this study, vancomycin and cefepime, the 2 wide spectrum, last resort and relatively expensive antibiotic products, did contribute the lion's share for the higher increase in glycopeptides (J01XA) and the fourth generation cephalosporins (J01DE) consumption, respectively. Both of these products are not, however, included in the list of essential medicines list for Ethiopia [38]. This makes it a worrisome development, to say the least. But given the indications of these two antibiotic products, particularly their common use for the treatment of infections caused by some resistant strains of bacteria, and the possibly increasing challenges related to antimicrobial resistance in Ethiopia, their increasing use even without being in the list of essential medicines may not have come as an unexpected. This is to say, given their indications for drug resistant infections, their increasing use may not be dismissed as 'irrational use' just at face value. To this end, vancomycin remains to be the drug of choice for the treatment of infections caused by Methicillin-Resistant-Staphylococcus Aureus (MRSA) including skin and soft tissue infectious and pneumonia [39]. Moreover, the fourth generation cephalosporins are used to treat infections caused by gram-negative bacteria which had already become resistant to other antibiotics [40]. This, therefore, can possibly explain the reason for the increasing use of these two classes of systemic antibiotics in Ethiopia, as documented in the current study.

What's more, perhaps as directly connected to the points discussed above, the decrease in the community-based consumption of the classes of systemic antibiotics like beta-lactamase-sensitive penicillins observed in the current study could be attributed to a practical reciprocation—or rather a pragmatic response—to the increase in the consumption of the fourth generation cephalosporins which are used for the treatment of infections caused by gram-negative bacteria. In line with the findings of the current study on this point, decreasing trends in consumption for antibiotic classes like amphenicols and beta-lactamase-sensitive penicillins were also reported elsewhere [41].

As to the most commonly utilized antibiotic products from among all the classes of systemic antibiotics (J01) investigated in the current study, all the nine antibiotic agents which constituted the common domain for the five years long study period (amoxicillin, doxycycline, ciprofloxacin, cloxacillin, metronidazole, trimethoprim, sulfamethoxazole, ampicillin and norfloxacillin) were also part of the systemic antibiotic products which constituted DU90% in a large tertiary care hospital in Ethiopia, as documented in our earlier study [19]. Proposed in the 1990s, DU90% is a simple index—an innovative tool by the time—used in drug utilization researches as a quality indicator of prescribing practices in healthcare institutions or beyond. By ranking the volume of drugs utilized as computed and compiled in terms of standardized measurement unit DDD, DU90% identifies the list of drugs accounting for 90% of the total drug consumption [35, 42]. Generally speaking, the shorter the list of the drug products and the most, if not all, of them being based on the official (institutional) clinical guidelines constituting the DU90%, the better the quality of prescribing practices [43].

Though commonly used as quality indicator in the assessment of healthcare facility-based prescribing practices, particularly as reference to institutional or official clinical guidelines or

local formularies, the concept of DU90% can also be appropriated to assess aggregate drug utilization trends at national and/or sub-national levels. And especially this can be more useful if done in reference to the official list of essential medicines for a given country or in continental sub-region. Such data can also provide invaluable tool to control or optimize expenditure on drugs and hence the overall budget for medications. In the current study, only 8 of the 13 antibiotic products (61.5%) which constituted DU90% during anyone of the year along the study period were from Ethiopia's official list of essential medicines. Antibiotic products like ciprofloxacin, norfloxacin, clarithromycin, tinidazole and azithromycin were part of the antibiotics constituting DU90% in one or more of the years during the study period (2016–2020), while not being included in the official list of essential medicines for Ethiopia. This also should certainly be a cause for huge concern to the country's health authorities.

In this study, community-based systemic antibiotic consumption in Ethiopia and Norway were also compared at the level of the chemical subgroups (ACT4) over 3 years period (2016–2018). For the 3 years period considered in the current study, community-based annual consumption of systemic antibiotics in Ethiopia were not quantitatively higher than those of Norway. However, trend in community-based consumption of systemic antibiotics over the study period showed an increase for Ethiopia while the opposite trend was observed for Norway. Here, the fact that community-based annual consumption of systemic antibiotics for Ethiopia during the 2016–2018 study period were lower than those of Norway may not, however, mean that Ethiopia had a better system of antimicrobial stewardship program than Norway to control and keep the rational use of antibiotics within check. In fact, quite the opposite could be true as far as robust system of antimicrobial stewardship program to ensure the rational use of antibiotics is concerned. That is to say, Norway has a far better developed and functioning system of antimicrobial stewardship which is exemplary even to the rest of the world [44]. The fact that a decreasing trend was observed in the community-based annual consumption of systemic antibiotics from 2016–2018 for Norway, as clearly documented in this study, corroborates this point. But in general, the reason for the quantitatively lower consumption of systemic antibiotics per year in Ethiopia during the study period (2016–2018) could rather be due to other factors related to the problems of affordability of essential medicines, particularly antibiotics, in the country's public health sector, as has been documented in our earlier study [45] and also elsewhere [46]. Besides, the lower consumption data could be related to other socio-cultural determinants like low healthcare services seeking behaviors of the general population living in the vast rural parts of Ethiopia [47–49]—of course to the detriment to the total health of the general population [50].

Beta-lactam antimicrobials, penicillins (J01C) constituted relatively higher percentages of community-based systemic antibiotics consumption for both Ethiopia and Norway. However, macrolides, lincosamides (J01F) and quinolones (J01M) constituted within the lower percentage ranges of the community-based systemic antibiotic consumption for Ethiopia and Norway, respectively, during the study period. Variations were observed between the proportions of the antibiotic classes constituting the higher and lower percentage consumption for both countries. Furthermore, a closer look at the rest of the data sets showed a characteristically heterogeneous community-based consumption of systemic antibiotic classes for both countries. These quantitative as well as qualitative differences in systemic antibiotics consumption between Ethiopia and Norway, as documented in this study, can be explained by various factors including differences in distribution of diseases (epidemiology), socio-economic determinants, differences in the structures of healthcare systems, established traditions in prescribing practices, and other factors which may push trends of antibiotic prescriptions, and therefore their consumption, towards an upward or downward spiral.

The major strength of this study is that it systematically documented—for the first time—the community-based annual consumption of systemic antibiotics in Ethiopia over five years period, using standard methodology and applying standardized measurement system (DID) to estimate the exposure of the general population to antibiotics in the country. It also offered in-depth analyses, using dis-aggregated data, on the changes and trends of community-based consumption of systemic antibiotics over five years in Ethiopia—including on comparative analysis of the Ethiopian data with that of Norway over a period of 3 years.

Its limitations however include its inability, due to lack of further dis-aggregated data, to capture seasonal variations in the community-based consumption of systemic antibiotics in Ethiopia during each of the five years that constituted the study period. Moreover, the study did not account for, also due to lack of data, the consumption of systemic antibiotics in the country's private health sector. In terms of the total number of healthcare facilities available in the country and hence their provision of healthcare services, the private sector in Ethiopia accounts for about 27% [51]. Assuming that the sector accounts for same proportional ratio in the community-based systemic antibiotics consumption in the country, this may therefore have had a downward bias–to that extent–on our estimates of the annual community-based consumption of systemic antibiotics in Ethiopia.

Moreover, it should also be stated here that challenges associated with the supply chain structures of these systemic antibiotics in particular and all other pharmaceuticals in general appear to strain their efficient distribution across all corners of Ethiopia's public health sector. Among the sources of such challenges could perhaps be the organizational and operational structures of EPSA itself. Ethiopia is officially a multinational federation [52] with 2 quasi-autonomous administrative city councils (Addis Ababa and Diredawa) accountable to the federal government and 9 autonomous member states of the federation (Tigray, Afar, Amhara, Oromia, Somali, Southern region, Harari, Benishangul Gumuz and Gambella). Each of these states got their own independent bureaucratic establishments in all sectors except defense and foreign affairs. Yet, a closer look at EPSA's supply chain structures to the branches and its so called "clusters" shows that, administratively, the centralized federal agency does not follow the country's federalist state structure. It therefore remains unclear as to how the agency harmonizes its operations with the bureaucracies of the regional states, under whose jurisdictions all public health facilities it stands to serve fall, to efficiently discharge its roles and responsibilities in the face of, arguably, the most contested and constrained fiscal federalism [53] in the Ethiopian federation.

## Conclusion

This study documented, for the first time, the community-based annual consumptions of systemic antibiotics (J01) over a period of five years (2016–2020) in Ethiopia. It also offered in-depth analyses on the trends of the qualitative as well as quantitative aspects, including the DU90%, of the community-based systemic antibiotics consumption over the study period. Accordingly, the study showed a significant increase (16.4%) in the consumption of systemic antibiotics over the five years period between 2016 and 2020 in Ethiopia—with a log-linear regression analysis showing an average annual increase in systemic antibiotics consumption by 3.4% ($R^2 = 0.89$). Wide spectrum antibiotics belonging to WHO's "watch and reserve group" like vancomycin and cefepime—from glycopeptides (J01XA) and the fourth generation cephalosporins (J01DE), respectively—showed staggering increases as high as 600–1300% in consumption during the study period. This should certainly be the cause of great concern for health authorities in Ethiopia. Moreover, comparative analysis of Ethiopia's data versus Norway's community-based systemic antibiotics consumption over 3 years period (2016–2018)

showed trends with opposite trajectories—increasing and decreasing trends observed for Ethiopia and Norway, respectively. Ethiopia should invest in policy actions to establish and institutionalize a robust program of antimicrobial stewardship.

## Supporting information

**S1 Fig. Log-linear regression model for systemic antibiotics consumption (DID) over a time period (2016–2020) in Ethiopia.**
(DOCX)

## Acknowledgments

Mr Gadissa Homa provided us extensive technical supports in the data collection phase of this study. We are extremely grateful to him. Dr Dereje Gudicha, a statistician at Wayne State University, Detroit, Michigan, helped us in the statistical analyses. We are thankful for his professional support.

## Author Contributions

**Conceptualization:** Girma Gutema.

**Data curation:** Girma Gutema.

**Formal analysis:** Girma Gutema, Seid Ali.

**Investigation:** Girma Gutema.

**Methodology:** Girma Gutema, Sultan Suleman.

**Writing – original draft:** Girma Gutema.

**Writing – review & editing:** Seid Ali, Sultan Suleman.

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
