## [Decision Letter · Decision Letter 0]

24 Feb 2021

PONE-D-20-39055

Trends of community-based systemic antibiotic consumption: Comparative analyses of data from Ethiopia and Norway calls for public health policy actions

PLOS ONE

Dear Dr. Gutema,

Thank you for submitting your manuscript to PLOS ONE. After careful consideration, we feel that it has merit but does not fully meet PLOS ONE’s publication criteria as it currently stands. Therefore, we invite you to submit a revised version of the manuscript that addresses the points raised during the review process.

We look forward to receiving your revised manuscript.

Kind regards,

Grzegorz Woźniakowski, Full professor, PhD, ScD

Academic Editor

PLOS ONE

Journal Requirements:

2. Please ensure that all conclusions meet PLOS ONE publication criterion number 4, which states that conclusions are presented in an appropriate fashion and are supported by the data (https://journals.plos.org/plosone/s/criteria-for-publication#loc-4). In particular, PLOS ONE does not publish policy papers, so please ensure that the policy implications of the findings are supported by the data.

Reviewers' comments:

Reviewer's Responses to Questions

**Comments to the Author**

1. Is the manuscript technically sound, and do the data support the conclusions?

Reviewer #1: Yes

2. Has the statistical analysis been performed appropriately and rigorously? 

Reviewer #1: Yes

3. Have the authors made all data underlying the findings in their manuscript fully available?

Reviewer #1: Yes

4. Is the manuscript presented in an intelligible fashion and written in standard English?

Reviewer #1: Yes

5. Review Comments to the Author

Reviewer #1: The article "Trends of community-based systemic antibiotic consumption: Comparative analyses of data from Ethiopia and Norway calls for public health policy actions" is interesting. Authors made a great work with collecting the data from Ethiopia I believe that it had to be very time consuming. There is a merit in the paper, however there are also some gaps that must be fulfilled/revised. From small things:

Pages 16-17: the font is bigger than in previous and later text.

Page 18 “[J01XA)” - wrong bracket

From more important things:

The introduction require data from Norway and Ethiopia about use of antibiotics (one paragraph). I notice that you mention that “Data on annual antibiotic consumptions in public health sectors are not publicly available for Ethiopia” but maybe there are some information about use of antibiotics in agriculture and in medicine, types of antibiotics or at least situation in Africa.

Results and Discussion should be written separately - this is my major concern. Discussion should be a separate paragraph and must be enhance in literature.

Sincerely,

Reviewer

6. PLOS authors have the option to publish the peer review history of their article (what does this mean?). If published, this will include your full peer review and any attached files.

Reviewer #1: No

---

## [Author Response · Author response to Decision Letter 0]

12 Mar 2021

Response to the reviewer 

First of all, we would like to thank the reviewer for his/her constructive comments on the manuscript and also for the bold appreciation and encouragement extended to us on the time-consuming job of manually collecting the data from Ethiopia, doing the calculations and the analyses that we presented in our study. 

Below, we give our responses/reflections on the reviewer’s comments: 

Reviewer’s comment: “Pages 16-17: the font is bigger than in previous and later text.”

Our response: The reviewer is right. We have accordingly corrected this in the revised manuscript. 

Reviewer’s comment: “Page 18 “[J01XA)” - wrong bracket”.

Our response: The reviewer is right. We have accordingly corrected this in the revised manuscript.

Reviewer’s comment: “From more important things...The introduction require data from Norway and Ethiopia about use of antibiotics (one paragraph). I notice that you mention that “Data on annual antibiotic consumptions in public health sectors are not publicly available for Ethiopia” but maybe there are some information about use of antibiotics in agriculture and in medicine, types of antibiotics or at least situation in Africa.”

Our responses/reflections: The reviewer is correct in recommending to include data about antibiotic use in Ethiopia and Norway in the introductory section of our manuscript – we noted that with compliments. We also noted reviewer’s acknowledgement about lack of studies on Ethiopia’s country-level consumption of antibiotics with compliments. There are however a few health facility-based studies published on antibiotic consumption and resistance, including one co-authored by one of the co-authors in this manuscript. We therefore have added one paragraph to the introductory part, as per the recommendation, citating those studies from Ethiopia and also the required country-level consumption data from Norway using references from 16 to 21. We therefore have corrected this, too, in the revised manuscript, accordingly. 

Reviewer’s comment: “Results and Discussion should be written separately - this is my major concern. Discussion should be a separate paragraph and must be enhance in literature.”

Our response: If taken with respect to PLOS ONE’s manuscript submission guidelines on “Manuscript Organization … Results, Discussion...”, which we did consult before submitting our article, the reviewer might not be correct because the submission guidelines offer an option to either put “Results and Discussion” together or present them separately. However, in line with the reviewer's recommendation, we have accordingly re-written the results and discussion separately in the revised version of our manuscript. 

Sincerely

Authors

---

## [Decision Letter · Decision Letter 1]

26 Apr 2021

Trends of community-based systemic antibiotic consumption: Comparative analyses of data from Ethiopia and Norway calls for public health policy actions

PONE-D-20-39055R1

Dear Dr. Gutema,

We’re pleased to inform you that your manuscript has been judged scientifically suitable for publication and will be formally accepted for publication once it meets all outstanding technical requirements.

Kind regards,

Grzegorz Woźniakowski, Full professor, PhD, ScD

Academic Editor

PLOS ONE

Additional Editor Comments (optional):

Reviewers' comments:

Reviewer's Responses to Questions

**Comments to the Author**

1. If the authors have adequately addressed your comments raised in a previous round of review and you feel that this manuscript is now acceptable for publication, you may indicate that here to bypass the “Comments to the Author” section, enter your conflict of interest statement in the “Confidential to Editor” section, and submit your "Accept" recommendation.

Reviewer #1: All comments have been addressed

2. Is the manuscript technically sound, and do the data support the conclusions?

Reviewer #1: Yes

3. Has the statistical analysis been performed appropriately and rigorously? 

Reviewer #1: Yes

4. Have the authors made all data underlying the findings in their manuscript fully available?

Reviewer #1: Yes

5. Is the manuscript presented in an intelligible fashion and written in standard English?

Reviewer #1: Yes

6. Review Comments to the Author

Reviewer #1: (No Response)

7. PLOS authors have the option to publish the peer review history of their article (what does this mean?). If published, this will include your full peer review and any attached files.

Reviewer #1: No

---

## [Editor Report · Acceptance letter]

6 May 2021

PONE-D-20-39055R1 

Trends of community-based systemic antibiotic consumption: Comparative analyses of data from Ethiopia and Norway calls for public health policy actions 

Dear Dr. Gutema:

I'm pleased to inform you that your manuscript has been deemed suitable for publication in PLOS ONE. Congratulations! Your manuscript is now with our production department. 

Kind regards, 

on behalf of

Prof. Grzegorz Woźniakowski 

Academic Editor

PLOS ONE